# Barriers and Facilitators to Weight and Lifestyle Management in Women with Polycystic Ovary Syndrome: General Practitioners’ Perspectives

**DOI:** 10.3390/nu11051024

**Published:** 2019-05-07

**Authors:** Alexis Arasu, Lisa J Moran, Tracy Robinson, Jacqueline Boyle, Siew Lim

**Affiliations:** Monash Centre for Health Research and Implementation, School of Public Health and Preventative Medicine, Monash University, Melbourne, VIC 3168, Australia; naara1@student.monash.edu (A.A.); lisa.moran@monash.edu (L.J.M.); Tracy.robinson@monash.edu (T.R.); Jacqueline.boyle@monash.edu (J.B.)

**Keywords:** PCOS, general practitioners, primary care, weight management, lifestyle management, implementation, guidelines, barriers, facilitators

## Abstract

Background: Weight and lifestyle management is advocated as the first-line treatment for polycystic ovary syndrome (PCOS) by evidence-based guidelines. Current literature describes both systems- and individual-related challenges that general practitioners (GPs) face when attempting to implement guideline recommendations for lifestyle management into clinical practice for the general population. The GPs’ perspective in relation to weight and lifestyle advice for PCOS has not been captured. Methods: Fifteen GPs were recruited to take part in semi-structured interviews. NVIVO software was used for qualitative analysis. Results: We report that GPs unanimously acknowledge the importance of weight and lifestyle management in PCOS. Practice was influenced by both systems-related and individual-related facilitators and barriers. Individual-related barriers include perceived lack of patient motivation for weight loss, time pressures, lack of financial reimbursement, and weight management being professionally unrewarding. System-related barriers include costs of accessing allied health professionals and unavailability of allied health professionals in certain locations. Individual-related facilitators include motivated patient subgroups such as those trying to get pregnant and specific communication techniques such as motivational interviewing. System-related facilitators include the GP’s role in chronic disease management. Conclusions: This study contributes to the understanding of barriers and facilitators that could be addressed to optimize weight and lifestyle management in women with PCOS in primary care.

## 1. Introduction

Polycystic ovary syndrome (PCOS) is a common and complex endocrine disorder affecting women of reproductive age with a prevalence of 8% to 13% [1]. It is associated with a range of features including hyperandrogenism, polycystic ovaries, and menstrual disturbances, which forms the Rotterdam diagnostic criteria [2]. Women can also experience additional complications with adverse reproductive, psychological, and cardio-metabolic outcomes [3]. Insulin resistance is key in the pathophysiology of PCOS and underpins the reproductive and metabolic presentations of PCOS including hyperandrogenism and ovarian dysfunction [4]. Insulin resistance is present in women with PCOS independent of their BMI [5,6]. Additionally, being overweight or obese is a well-recognized, significant feature and complication of PCOS for many women [7,8,9,10]. The Australian National Women’s Health policy identifies PCOS as a key obesity-related reproductive problem and a significant health burden [11]. Insulin resistance and its complications are worsened by increasing weight, whilst appropriate lifestyle interventions and weight loss can reduce insulin resistance [12]. The 2018 updated PCOS evidence-based guidelines recommend lifestyle management to be first-line treatment, regardless of presenting symptoms [13]. Lifestyle management can be defined as diet, exercise, and behavioral interventions. Weight management through lifestyle management is also crucial and encapsulates weight loss, weight maintenance, or prevention of weight gain. The benefits of lifestyle management have been demonstrated by a large body of literature to show that weight loss as little as 5% to 10% improves the reproductive, endocrine, cardio-metabolic, and psychological features of PCOS [4,13,14]. Additionally, lifestyle management independent of weight loss has independent, positive benefits on insulin resistance and the features of PCOS [3,14].

Given the complexity of PCOS, its management requires a multi-disciplinary team that includes general practitioners (GPs), endocrinologists, gynecologists, and allied health professionals. GPs are often the first port of call for patients and have access to over 80% of the population [15,16,17]. GPs can maintain long-term relationships with their patients and have a holistic perspective on patient care [15,17,18,19,20]. GPs have the ideal platform to provide lifestyle management. Moreover, patient desire for GP involvement in lifestyle management has been expressed [15,21,22]. Despite this, literature reports in the general population that GPs may not be fulfilling this role because of system-related and individual-related factors. These may include poor dissemination of guideline-based information to GPs, lack of funding, competing general practice interests, and time and pressure demands [21,23,24,25,26]. However, the perspective of GPs in providing weight and lifestyle management for women with PCOS has not been addressed in the literature. This is crucial given the common prevalence of PCOS, the key association of weight with worsening the presentation and prevalence of PCOS, and the key role of GPs in the care of women with PCOS. To improve the quality of care received by women with PCOS, documenting GPs’ perspectives can identify gaps in knowledge translation and provide specific recommendations for future implementations. We aim to understand the knowledge and practice of GPs on weight and lifestyle management in women with PCOS, specifically, barriers and facilitators for GPs in meeting the current recommendations for weight and lifestyle management.

## 2. Materials and Methods

This study used a qualitative design to investigate broad contextual issues influencing the translation of lifestyle interventions from clinical guidelines into primary care in women with PCOS. GPs were recruited to take part in semi-structured interviews, conducted face-to-face or over the phone. Inclusion criteria stipulated five participants working in rural or regional areas, five having a large proportion (30%–40%) of a client base from cultural and linguistically diverse (CALD) backgrounds, and five working with individuals from the general population. No participants were excluded as they were purposively selected to fulfill the inclusion criteria. Qualitative research is dictated by data saturation, and this can be achieved through smaller sample sizes. This was previously demonstrated in a qualitative GP and weight intervention paper with a sample size of 12 [27]. A total of 23 GPs were purposively contacted, utilizing existing contacts and broad internet searches; five of those GPs agreed to participate. The remainder of participant recruitment was facilitated through snowball sampling. All participants provided informed consent. The protocol was approved by the Monash Health Human Research Ethics Committee.

The interview schedule utilized a funneling method, with inherent flexibility for the researcher to either elaborate or clarify certain responses. The schedule ensured questions were open-ended and exploratory in nature. Basic demographic data including gender, age, years of practice, frequency of PCOS encounters, and place of practice were collected. The interviews explored current perceptions on the knowledge and practice of weight and lifestyle management in women with PCOS. Specifically, each practitioner described the extent to which they were able to provide weight and lifestyle management for women with PCOS, barriers and facilitators influencing practice, and suggested improvements to address these barriers.

The semi-structured interviews were conducted by one researcher (AA, a female medical student), allowing for continual cross-checking of themes and assessing when data saturation was achieved, with no new ideas emerging. Thematic saturation was achieved within 15 participants.

Interviews were audio recorded and conducted in the GP consulting rooms or over the phone. Duration of interviews was between 20–30 min. De-identified audio recordings were transcribed verbatim using an independent transcribing service.

Transcripts were thematically analyzed and coded using NVIVO Qualitative data analysis software; QRS International Pty Ltd. Version 12, 2018, Melbourne, Australia. Using an inductive process, codes were generated according to the emergent patterns from participant discussions [28,29]. The iterative process of mind-mapping themes and subthemes was used until categories were identified. A random sample of 10% was coded and thematically analyzed independently by a second researcher (SL). The resulting codes and themes were compared between the two researchers (SL, AA), and discrepancies were discussed and reconciled.

## 3. Results

Baseline characteristics from all 15 participants were as shown in Table 1. Knowledge, barriers, and facilitators to weight and lifestyle management in PCOS are described in Table 2.

### 3.1. Knowledge on the Importance of Weight and Lifestyle Management and the Use of Guidelines

Participants identified the importance of weight and lifestyle management and its critical role as first-line treatment for women with PCOS.

“I’d say it’s the biggest role. So, first-line treatment would be lifestyle change and looking at weight reduction and increasing exercise and activity.”—Participant 8, rural

Discrepancies arose in the routine use of guidelines in clinical practice. Many participants stated their familiarity with guidelines, the ease of use of guidelines, and their reliance of guidelines to aid clinical decision making. However, others cited limitations pertaining to accessing guidelines and the length of the guidelines, which impeded practical use within primary care especially during a patient consult.

“You give me 300 pages of guidelines I’m not going to read it.”—Participant 6, Rural

Some GPs felt inundated with information from guidelines, and at times they felt their capacity was limited to keep updated with all conditions within the scope of primary care. Participants also expressed preference for summarized forms of information such as flowcharts.

“We are overrun with guidelines There are literally just guidelines pouring out left, right and center, and we deal with every condition, every age group.”—Participant 4, non-CALD

### 3.2. Barriers to Weight and Lifestyle Management

Participants reported a number of systems-related and individual-related barriers and facilitators relating to implementation of weight and lifestyle management for PCOS in clinical practice.

#### 3.2.1. System-Related Factors

In terms of the barriers and facilitators within the primary care system, participants value, and rely on, allied health services to manage lifestyle in PCOS, particularly the use of dietitians. Despite this, GPs working in rural areas face issues in terms of the availability of these services. A qualitative difference was observed in rural participants, who uniquely expressed difficulty getting access to allied health services compared to urban participants.

“There are huge barriers in terms of regional areas…some of the places if I’m referring to see a dietitian… it may be three-plus months before they could potentially get into see somebody.”—Participant 9, Rural

The perceived cost barrier, however, was present in all subgroups, regardless of the demographic of patient seen. The limited number of subsidized visits to allied health services through the chronic disease management plan (CDM) plan was also seen as a financial limitation to accessing allied health services.

“There’s an out of pocket cost for the patients and sometimes they just can’t afford that out of pocket cost. Even though we as GPs bulk bill them, our allied health colleagues do not.”—Participant 6, Rural

#### 3.2.2. Individual-Related Factors

Participants reported barriers such as weight and lifestyle management being professionally unrewarding, lack of time during consultation, and reimbursement issues relating to length of consultation.

The concept of weight and lifestyle management being professionally unrewarding was driven by two major facets: the first was the nature of lifestyle management interventions taking time to show results, the second was the results of preventative care being relatively small and more difficult to appreciate.

“It’s difficult to measure that sort of success. I think with lifestyle conditions, its sometimes difficult to manage because you don’t feel necessarily a great sense of satisfaction because it’s very difficult to get good results. So, it’s not like. I’ve seen a grossly overweight woman suddenly... you know, in two years’ time, becoming lean. It’s unrewarding in the long run when they come back and they you weigh them, and they haven’t really lost much weight, or any weight at all.” —Participant 5, non-CALD

Lifestyle counselling requires longer consultations, to which participants reported lack of time as a barrier. Participants had average appointment times between 10–15 min and were bulk billed, which was perceived to not facilitate lifestyle counselling of this nature.

“I do my best, but obviously I can’t spend a lot of time, talking about diet and give explanations. I just outline, in general, what they have to do.”—Participant 14, CALD

Time constraint issues were more prominent when viewed in conjunction with the perceived lack of reimbursement, with the perception that the Medicare structure financially penalized longer consultations.

“The way the Medicare schedule is certainly set up is you’re rewarded for seeing people for short periods of time.”—Participant 11, CALD

“I don’t think it rewards those GP’s who are really thorough and want to try to address and cover all the issues that are there. I think, unfortunately, those doctors are penalised.”—Participant 9, Rural

Additionally, some participants perceived that some patients were hindered by their own lack of motivation or willingness to accept the help that was offered.

“It’s patient-driven and it’s always hard to get people to actually go and do the resource. We’re really good at making up new resources, but you’ve got to get the person who’s got the metabolic problem to actually want to use the resource.”—Participant 2, Non-CALD

GPs expressed perceptions that some patients were looking for a ‘quick fix’, which seemed more realistic to them than trying to make changes to their lifestyle.

“No one wants to hear that if you exercise and eat healthily your acne will get better in maybe a year’s time. People want a quick fix for that.”—Participant 3, Non-CALD

Some GPs also expressed difficulties in approaching discussions surrounding weight-related issue with patients.

"It is a really hard thing to discuss to people, you know. It’s hard because it is so emotionally laden.”—Participant 3 Non-CALD

### 3.3. Facilitators of Weight and Lifestyle Management

#### System-Related Factors

Participants agreed on the importance of allied health professionals, specifically dietitians, in providing lifestyle management. Few participants also mentioned exercise physiologists and psychologists as forming part of the multi-disciplinary team. The CDM plans, which enable GPs to allocate five subsidized visits to allied health professionals for patients with chronic disease, were seen as a facilitator to lifestyle management in PCOS. 

“There are some systems in place, like, chronic disease management plans that we can use to refer patients off to dietitians or exercise physiologists so I guess that’s one positive thing that’s out there.”—Participant 3, Non-CALD

Participants believe the role of the GP to be an important facilitator, as they can act as a care coordinator and provide continuity of care.

“I think a GP is a great person to help coordinate that care, because they provide holistic care.”—Participant 3 Non-CALD

Participants also acknowledged the importance of communication and counselling skills including motivation interviewing. A number of participants felt they had these skills, which facilitated their ability to implement lifestyle management in women with PCOS.

“I think that’s where motivational interviewing comes in, and that’s a skill that GPs often do have, which is a way of discussing with patients, and eliciting their own motivation for activity.”—Participant 5, Non-CALD

The presence of immediate health concerns, such as infertility, were effective motivators to lifestyle changes in patients. Participants cited that the greater the motivating factor in the context of each individual patient, the greater the engagement. Similarly, those that did not have a set goal were far less likely to commit.

“That’s a big motivating factor for women. You know, one of the primary reasons, I think, they come and see you, as a GP, is because they’re trying to get pregnant.”—Participant 9, Rural

## 4. Discussion

In this novel qualitative study, we explored the perspective of GPs on perceived facilitators and barriers in the knowledge and practice of the implementation of lifestyle management for PCOS in primary care. We report that GPs recognize and understand the importance of lifestyle management and its role as a first-line treatment for PCOS. Barriers relate to lack of funding for longer GP consultations, time constraints, patient motivation, and physical and financial accessibility to allied health professionals. Facilitators relate to ongoing, holistic GP–patient relationships, specific GP communication skills, including motivational interviewing, and allied health referrals. We also explored areas where changes may improve care of women with PCOS.

We report that GPs in this study felt confident and knowledgeable of the guideline lifestyle recommendations for PCOS as well as how best to deliver counselling for lifestyle-related disease. Despite this, previous studies regarding lifestyle management for the general population suggested that much of the lifestyle counselling provided by GPs was generic [23], and GPs often lacked confidence or had limited specific training to help support necessary sustainable behavioral changes [18,25,30,31,32]. Clinical practice guidelines remain to be one of the most important reference tools used by GPs in our study, as consistent with previous studies [33]. Whilst many demonstrated their familiarity and ease of using guidelines, some did express feeling overwhelmed [33]. Length, depth, and variety of clinical guidelines can reduce the GP’s ability to stay updated, which may contribute to a pattern of low adherence to guidelines in general practice [33,34,35,36], thus explaining the predominance of giving more generic lifestyle related information to patients. As previously reported by Taba et al., we found strong consensus supporting summarized, printed forms of guidelines such as one-page snapshots and flow charts [33]. These are currently available with the PCOS guidelines [3]. Greater use of these resources may allow for greater awareness and routine implementation of guideline recommendations.

In their randomized controlled trial, Pritchard et al. reported that collaboration between GPs and dietitians led to the most successful weight loss and health promotion when compared to individual practitioner attempts. This was consistent with other reports that combined care between GPs and allied health professionals, such as dietitians, resulting in greater motivation from patients in lifestyle interventions [30,37,38,39]. Our findings indicate routine referral to dietitians supports greater uptake of lifestyle management. This contrasts with existing literature on the practice habits from GPs, suggesting limited referrals to dietitians from GPs [34,37,40,41,42]. Patient reluctance to visit dietitians is similarly documented [22,38,40,43,44]. In Australia, CDM plans reduce the complexity of allied health referrals, as it allows GPs to organize five subsidized allied health care visits for patients with chronic disease. However, the inadequacy of five visits was expressed, and in consensus with previous research, patients will likely incur out of pocket costs that inhibit engagement [45]. Increasing the provision of greater financial support for allied health professionals may increase utilization from patients. Those living in rural and remote areas face an additional challenge in procurement of allied health services because of the lack of availability and physical accessibility of allied health professionals; thus, the lack of access to allied health services for women with PCOS may lead to poorer outcomes compared to their metropolitan counterparts. Proximity to capital cities positively influences health outcomes. Those living rurally experience poorer nutrition, lower levels of physical activity, and higher obesity rates, which contributes to urban–rural health disparity [46]. Building strong primary care foundations, with reliable access to allied health professionals to support early lifestyle interventions, may help to address this disparity. Currently, the demand for allied health services far exceeds the supply in rural and remote areas [47]. Our findings suggest that poor distribution of allied health services in rural and remote areas is a potential barrier to equitable translation of the guidelines in implementing lifestyle management in vulnerable populations such as those in living rural and remote areas.

Whilst GPs may grant referrals to allied health professionals, prescription of lifestyle advice is often facilitated by the GP. In the general population, time constraints and lack of reimbursement were prominent themes recurrent in previous literature, which prevented GPs from efficiently providing lifestyle management. We report for the first time that similar barriers exist in general practice for PCOS [21,23,24,25,26]. Effective lifestyle counselling requires thorough and longer consultations. This can be challenging for GPs, as they often encounter multiple health concerns within a consultation [48,49,50,51], with a tendency for lifestyle management to often be overlooked in this context [21]. Within the Australian health system, longer consultations do not carry the same financial reimbursement compared with shorter consultations. Thus, GPs in our study that were thorough and spent longer with their patients felt penalized under the current Medicare schedule. However, some were able to circumvent this problem by scheduling shorter, more frequent follow up appointments [52]. This also enables closer monitoring and support of progress [49,50]. However, this may not fully address the need for longer consultations for lifestyle management, particularly in the initial consult. Hence, lack of time remains one of the primary barriers to lifestyle management by GPs [18,19,21,25,30,32,39,43,52,53]. A system change may be required to reward longer, and more thorough GP consultations, given the evidence in support of greater patient satisfaction and preventative care from longer consultation length [54]. Additionally, as previously mentioned, greater reliance and engagement of allied health professionals may reduce this time burden on GPs [52].

Lifestyle management for the general population has often been characterized to be unrewarding by GPs; the reasons are multifaceted, and our current findings in PCOS are in support with a large body of literature [25,30,34,55,56,57,58,59]. Problems arise with issues of lifestyle interventions, as they are perceived as offering slow return, have low tangibility, and patient motivation is viewed as an additional barrier [48,50,60]. Unrealistic expectations regarding lifestyle changes from both practitioner and patient may lead to feelings of failure [56]. However, modest weight reductions of 5%–10% improve the endocrine, reproductive, metabolic, and psychological features of PCOS [14]. Weight gain prevention is also important, as women with PCOS have been shown to have greater longitudinal weight gain. This could also prevent long-term adverse sequelae such as diabetes and cardiac complications [9]. Creating realistic healthy target weights are key. Changes to perceptions, and advocating for smaller incremental improvements as a success, may allow GPs to develop a greater sense of satisfaction [48]. This may be facilitated by increased education and awareness of the management outcomes for PCOS. Our findings suggest that issues with patient motivation can also contribute to lifestyle management being perceived as unrewarding by GPs, as consistent with previous research in the general population [48,50,60]. There is a perception that patient motivation is deeply rooted and difficult to modify [61]. Many health professionals involved in general lifestyle counselling perceive that patients must be responsible for making changes and attempts to intervene are often halted by the patient’s unwillingness to change [25,48,56]. A shift to understanding how patient-specific motivators could be modified may be key, which is the main principle underpinning motivational interviewing [45]. Its employment for changing lifestyle behaviors such as smoking cessation and alcohol dependence has been shown to be effective [45]. There is burgeoning research to suggest the value of motivational interviewing in the context of lifestyle interventions for women with PCOS [62,63].

Discussing lifestyle-related issues, such as confronting overweight patients, has often been met with hesitancy by doctors [37,38,39]. We encountered few GPs that shared these sentiments. In accordance with previous literature in the general population, GPs convey difficulties in facilitating discussions of this nature because of a fear of offending patients [38]. Despite this, open, honest conversations and development of specific communication techniques over time can increase effectiveness of delivering lifestyle advice and overcoming these challenges [40]. Issues with GP confidence in discussing lifestyle-related counselling for the general population has surfaced in previous literature [18,25,30,31,32]. Our finding suggests otherwise, as we report that overall, GPs in our study displayed high confidence in managing PCOS and discussing lifestyle counselling. Notably, we found that participants seldom referred their patients to specialists, except in the context of infertility treatment. This highlights that a large proportion of women with PCOS may be predominantly engaged in the primary care setting. It is important to note, however, that our results may have been influenced by the nature of participant sampling. Those willing to participate in interviews may have done so given their greater interest in PCOS, and they may have had more experience managing women with PCOS. Thus, results may not necessarily reflect the general population. Additionally, some of our participates were affiliated with university teaching units and, therefore, had greater baseline knowledge and a willingness to engage with research.

## 5. Conclusions

We report that GPs are knowledgeable and reference current guidelines when providing weight and lifestyle management for women with PCOS. Facilitators of weight and lifestyle management include referrals to allied health professionals, motivating patients with specific health concerns, and the ongoing, holistic GP–patient relationship. Practice barriers were related to funding and structural issues preventing adequate access to allied health professionals, time constraints, lack of reimbursement for long consultations, and issues with motivation from both GPs and patients alike.

These factors provide insight to potential avenues, which may require a combination of system and individual interventions, to strengthen and optimize health outcomes for women with PCOS and lifestyle-related disease.

Recommendations include brief snapshot guidelines, developed specifically for primary care, that could be more easily integrated into GP consultations; this would ensure the highest level of evidence-based practice to promote greater use of guidelines by GPs.

Stronger and more incentivized referral pathways for GPs to recruit allied health professionals is necessary to support women with PCOS, with a specific focus on GPs working in rural and remote areas. This research is the first to report on GP-perceived facilitators and barriers in supporting women with PCOS in weight and lifestyle management. Future research should be directed towards assessing GP perspectives regarding diagnosis and management for women and understanding the entire model of care concerning weight and lifestyle management for women with PCOS.

Although this study is qualitative, and broad generalizations of findings may be limited, this research provides crucial understanding from the GP perspective in relation to providing weight and lifestyle management for women with PCOS, and, thereby, it contributes to a wider body of research examining weight and lifestyle management in primary care.

## Figures and Tables

**Table 1 nutrients-11-01024-t001:** Baseline characteristics of participants.

Characteristics	Responses	*N* (%)
Gender	Men	4 (26)
	Women	11 (74)
Geographical location	Metropolitan	10 (67)
	Rural	5 (33)
Number of years in practice	1–5 years	1 (7)
	6–10 years	5 (33)
	11–15 years	2 (13)
	15–20 years	3 (20)
	21+ years	4 (26)
Estimate of frequency of PCOS patients encountered	Weekly basis	9 (60)
	Monthly basis	1 (7)
	Yearly basis	5 (33)
Patient demographic	CALD * (at least 30–40% of patients from CALD backgrounds)	5 (33)
	Non-CALD (<30–40% of patients from CALD backgrounds)	6 (40)
	Aboriginal and Torres Strait Islanders	4 (27)

* CALD: Cultural and linguistically diverse. PCOS: Polycystic ovary syndrome.

**Table 2 nutrients-11-01024-t002:** Facilitators and barriers on knowledge and practice in lifestyle management of PCOS by GPs.

Categories	Sub-Categories	Themes
**Knowledge**		Importance of lifestyle management Practicality of guidelines in clinical practice
**Practice barriers**	System-related factors	Physical access to allied health professionals Financial access to allied health professionals
	Individual-related factors	Perceived lack of patient motivation for weight management Time constraints Lack of financial reimbursement Professionally unrewarding
**Practice facilitators**	System-related factors	Chronic disease management plans The general practitioner’s role in chronic disease management
	Individual-related factors	Perceived positive patient motivators Communication techniques/motivational interviewing

PCOS: Polycystic ovary syndrome.

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
