# Peer review of "Barriers and Facilitators to Weight and Lifestyle Management in Women with Polycystic Ovary Syndrome: General Practitioners’ Perspectives"

_nutrients, 2019, doi:10.3390/nu11051024_

Round 1
Reviewer 1 Report
The paper is well written and the methods are interesting. The difficulty of the general pratictioner in satisfying, in its clinical practice, the multiple guidelines of medicine is strongly noted. The questions asked and the answers obtained by the general practictioners are interesting for those of us who are specialists.
Probably a summary table could help the reader.
Author Response
Thank you for your comments. To clarify - in reference to "The questions asked and the answers obtained by the general practictioners are interesting for those of us who are specialists" - is this a general statement, or indicating they would like to see the questions and answers? If it is the latter, we would be happy to attach the interview schedule however as they were semi-structured interviews, the conversations were more fluid and dynamic.
Regarding
the summary table, please refer to Table 2 which summaries the main
categories and themes that arose from the interviews.

Reviewer 2 Report
In this manuscript, Arasu et al. describes the barriers and facilitators to general practitioners (GP) in Australia for the implementation of lifestyle management in women with PCOS using qualitative approach after semi-structural interviews of 15 GP. They favored a purposive sampling, consisting of rural and regional GP through “existing” contacts and snowball sampling. The authors report in this sampling that GP were in general aware of PCOS guidelines and importance of weight management in this population, were confident on their skills to manage lifestyle habits, but time-consuming appointments, lack of financial incentives and access to allied-professional were the main barriers for implementation.
Whereas the authors address a very important question to determine future actions to favor the implementation of weight loss interventions, my major concerns on this study are on the possibility of generalisation of these findings to other health systems and GP organisations, that seems very limited. Overall, the manuscript would benefit also from a better justification and report of the methodological choices.
MAJOR
1. Method
The choice of sampling needs to be justified more clearly in order to establish the generalization of the results. The sampling inclusion/exclusion criteria for the GP are not described by the authors. The justification for the proposed purposive sampling could be more explicit.
There is no details on the intended sample size and justification or the targeted sample source
A third of the GP interviewed had encounter with PCOS patients only yearly; making the choice of inclusion of this large proportion of GP without significant practice in the field of interest questionable.
2. Results:
The numbers of GP screened to participate/ GP included/excluded or refusing to participate should be reported. Reasons for exclusion should be reported if any.
3. Discussion:
Line 262-264. Discussion on rural vs urban GP barriers discussed but not reported in the results.
MINOR
1- Introduction; Line 39-40, Litterature suggests that obese women with PCOS are more likely to consult for their symptoms and have a diagnosis of PCOS, but are not necessarly more obese or overweight in unbiased sample (Ezeh U et al., J Clin Endocrinol Metab. 2013 Jun;98(6):E1088-9 al.)
2-Line 107- Barrier and facilitators from interviews- table 2 and not 1.
3-The title of table 2 could reflect that it consists of the barriers and facilitators resulting from GP interviews.
Author Response
We thank the reviewers for their helpful and constructive comments. Please see below our responses to the comments.
Reviewer feedback | Relevant amendments and explanations |
The choice of sampling needs to be justified more clearly in order to establish the generalization of the results. The sampling inclusion/exclusion criteria for the GP are not described by the authors. The justification for the proposed purposive sampling could be more explicit.
There is no details on the intended sample size and justification or the targeted sample source
The numbers of GP screened to participate/ GP included/excluded or refusing to participate should be reported. Reasons for exclusion should be reported if any.
| Thank you for your feedback, we have now included more information on the sampling, inclusion/exclusion criteria and agree a stronger rational was needed. (Pg 2, Lines 76-83) “No participants were excluded as they were purposively selected to fulfil the inclusion criteria. Qualitative research is dictated by data saturation, and this can be achieved through smaller sample sizes. This is previously demonstrated in a qualitative GP and weight intervention paper with a sample size of 12 (27). A total of 23 GPs were purposively contacted, utilising existing contacts and broad internet searches; five of those GPs agreed to participate. The remainder of participant recruitment was facilitated through snowball sampling.” |
A third of the GP interviewed had encounter with PCOS patients only yearly; making the choice of inclusion of this large proportion of GP without significant practice in the field of interest questionable.
| Our current sampling has broadly captured GP interactions with PCOS patients. It was important for us to recognise that many GPs in the population may have to provide management for PCOS patients, despite their encounter rates being low. PCOS only affects 8-13% of women of reproductive age. The inclusion of those who encounter women with PCOS only yearly provide a balanced view by including GPs with varying degree of experience in PCOS. The knowledge and practice habits imparted from GPs that do not encounter PCOS patients as frequently was seen just as valuable for our research. Given that this is a qualitative paper, we do appreciate that results are not broadly transferrable, but it does provide a snapshot of results. |
Line 262-264. Discussion on rural vs urban GP barriers discussed but not reported in the results. | This notion is discussed in the results section as a major qualitative difference which emerged between rural vs urban GPs relating to physical access to allied health professionals which was uniquely experienced in rural areas; (Pg 4, Lines 140-145)
It has been reworded to further highlight rural vs urban difference: (lines 140-145) “Despite this, GPs working in rural areas face issues in terms of the availability of these services. A qualitative difference was observed in the rural participants, who uniquely expressed difficulty getting access to allied health services compared to urban participants.” |
Introduction; Line 39-40, Litterature suggests that obese women with PCOS are more likely to consult for their symptoms and have a diagnosis of PCOS, but are not necessarly more obese or overweight in unbiased sample (Ezeh U et al., J Clin Endocrinol Metab. 2013 Jun;98(6):E1088-9 al.) | Thank you for bringing this to our attention – we have amended this to ensure that we acknowledge that not all women with PCOS are obese/overweight. The sentence has been amended to: Pg 1, Lines 39-40. “Additionally, being overweight or obese is a well-recognised, significant feature and complication of PCOS for many women.” |
Line 107- Barrier and facilitators from interviews- table 2 and not 1. | Thank you for sighting this error: Changed from Table 1 Ă Table 2 (pg 4 line 113) |
The title of table 2 could reflect that it consists of the barriers and facilitators resulting from GP interviews. | Table 2 title has been changed to “Facilitators and barriers on knowledge and practice in lifestyle management of PCOS by GPs” (pg 3 line 110) |
My major concerns on this study are on the possibility of generalisation of these findings to other health systems and GP organisations, that seems very limited. | Thank you for raising this point. The conclusion has been revised to include our acknowledgements that being a qualitative paper, broad generalisations from our research is limited, however it addresses important and critical areas of GP-provided care for women with PCOS, and by extension, challenges with weight loss and lifestyle interventions in primary care. Thus, addressing these concerns, may reverberate through many other lifestyle related conditions. Pg 9, Lines 336-339: “Although this study is qualitative, and broad generalisation of findings may be limited, this research provides crucial understanding from the GP perspectives in relation to providing weight and lifestyle management for women with PCOS and thereby contributes to a wider body of research examining weight and lifestyle management in primary care.”
|
Round 2
Reviewer 2 Report
All my comments have been addressed.